# Systematic review and network meta-analysis with individual participant data on cord management at preterm birth (iCOMP): study protocol

Anna Lene Seidler ,[1] Lelia Duley,[2] Anup C Katheria,[3] Catalina De Paco Matallana,[4] Eugene Dempsey,[5] Heike Rabe,[6] John Kattwinkel,[7] Judith Mercer,[8] Justin Josephsen,[9] Karen Fairchild,[7] Ola Andersson,[10] Shigeharu Hosono,[11] Venkataseshan Sundaram ,[12] Vikram Datta,[13] Walid El-Naggar,[14] William Tarnow-Mordi,[1] Thomas Debray,[15] Stuart B Hooper,[16] Martin Kluckow,[17] Graeme Polglase,[16] Peter G Davis ,[18] Alan Montgomery,[2] Kylie E Hunter ,[1] Angie Barba,[1] John Simes,[1] Lisa Askie,[1] On behalf of the iCOMP Collaboration

For numbered affiliations see end of article.

**Correspondence to**
Anna Lene Seidler;
lene.seidler@ctc.usyd.edu.au

## ABSTRACT

**Introduction** Timing of cord clamping and other cord management strategies may improve outcomes at preterm birth. However, it is unclear whether benefits apply to all preterm subgroups. Previous and current trials compare various policies, including time-based or physiology-based deferred cord clamping, and cord milking. Individual participant data (IPD) enable exploration of different strategies within subgroups. Network meta-analysis (NMA) enables comparison and ranking of all available interventions using a combination of direct and indirect comparisons.

**Objectives** (1) To evaluate the effectiveness of cord management strategies for preterm infants on neonatal mortality and morbidity overall and for different participant characteristics using IPD meta-analysis. (2) To evaluate and rank the effect of different cord management strategies for preterm births on mortality and other key outcomes using NMA.

**Methods and analysis** Systematic searches of Medline, Embase, clinical trial registries, and other sources for all ongoing and completed randomised controlled trials comparing cord management strategies at preterm birth (before 37 weeks' gestation) have been completed up to 13 February 2019, but will be updated regularly to include additional trials. IPD will be sought for all trials; aggregate summary data will be included where IPD are unavailable. First, deferred clamping and cord milking will be compared with immediate clamping in pairwise IPD meta-analyses. The primary outcome will be death prior to hospital discharge. Effect differences will be explored for prespecified participant subgroups. Second, all identified cord management strategies will be compared and ranked in an IPD NMA for the primary outcome and the key secondary outcomes. Treatment effect differences by participant characteristics will be identified. Inconsistency and heterogeneity will be explored.

**Ethics and dissemination** Ethics approval for this project has been granted by the University of Sydney

## Strengths and limitations of this study

► This will be the most comprehensive review to date of interventions for umbilical cord management in preterm infants and the findings will be highly relevant to clinicians and guideline developers.

► The use of individual participant data will allow assessment of the best treatment option for key subgroups of participants.

► Network meta-analysis will enable the comparison and ranking of all available treatment options using direct and indirect evidence.

► For some of the trials it will not be possible to obtain individual participant data, so published aggregate results will be used instead.

► Risk of bias in the primary trials will be assessed using Cochrane criteria, and certainty of evidence for the meta-analyses will be appraised using the GRADE (Grading of Recommendations Assessment, Development and Evaluation) approach for the pairwise comparisons, and the CINeMA (Confidence in Network Meta-Analysis) approach for the network meta-analysis.

Human Research Ethics Committee (2018/886). Results will be relevant to clinicians, guideline developers and policy-makers, and will be disseminated via publications, presentations and media releases.

**Registration number** Australian New Zealand Clinical Trials Registry (ANZCTR) (ACTRN12619001305112) and International Prospective Register of Systematic Reviews (PROSPERO, CRD42019136640).

## INTRODUCTION

Currently over 15 million babies are born preterm annually and this number is rising.[1–3] Of these, 1.1 million die, and the morbidity

BMJ

and healthcare costs among survivors and their families are high, with preterm survivors having an increased risk of cognitive, developmental and behavioural difficulties, and chronic ill health.[4–9] Hence, even modest improvements in outcomes of preterm birth would substantially benefit the children, their families and also health services. In uncompromised babies, deferring cord clamping has been shown to be beneficial and is now used in routine practice.[10] However, it is unclear whether these benefits apply to preterm babies who usually receive immediate neonatal care, and whether any benefits outweigh potential harms. In addition, there are multiple competing cord management strategies, such as clamping the cord at different times or milking the cord, and considerations of the infant's respiratory status, and it is currently unknown which strategy yields the best balance of benefits and harms.

### Current approaches to cord clamping

One potential mechanism of deferring umbilical cord clamping is a net transfer of blood from the placenta to the baby known as 'placental transfusion'. If the cord is not clamped at birth immediately, blood flow between the placenta and the baby may continue for up to 5 min in term infants.[11–13] For preterm births, blood flow may continue for longer,[14] since a greater proportion of fetoplacental circulating blood volume is still in the placenta.[15] This has led to time-based approaches to deferring cord clamping that have been shown to increase peak haematocrit and reduce the need for blood transfusions.[16] Yet, recent findings suggest that placental transfusion does not always occur—blood flow may continue without any net transfer, and sometimes net transfer may be to the placenta.[17] Initial neonatal care and stabilisation traditionally takes place on a resuscitation platform at the side of the room or in an adjacent room. Deferred cord clamping is thus often associated with a delay in neonatal care and this has led to concerns including delayed resuscitation and hypothermia[18] particularly for very preterm infants and infants assessed as requiring resuscitation. An alternate emerging strategy is to provide immediate neonatal care with the cord intact beside the woman using a mobile resuscitation trolley or on the mother's leg.[19–23]

Another potential mechanism of deferred clamping is allowing time for the infant to establish spontaneous breathing while still placentally supported. Immediate cord clamping before the infant has established breathing may be harmful since it can lead to large fluctuations in blood pressure, a period of hypoxia and restricted cardiac function.[24] Animal and pilot human studies suggest that breathing and lung aeration before cord clamping can improve cardiovascular stability and oxygenation, and reduce intraventricular haemorrhage and infant mortality.[25–28] They also suggest that initial respiratory support before clamping the cord can improve cerebral oxygenation and blood pressure, and reduce cerebrovascular impairment compared with immediate cord clamping.[29 30] This evidence has led to the rise of 'physiological cord clamping' which defers clamping until after the onset of breathing. Yet, onset of breathing is not always easy to determine without the assistance of video or extra equipment, while timing to cord clamping can be easily measured. In an earlier study,[31] time of onset of breathing in preterm infants receiving gentle stimulation was related to time after birth—within a minute over 90% of preterm infants had begun spontaneous breathing.

Cord milking or stripping (pinching the umbilical cord close to the mother and moving the fingers towards the infant) may be a way to increase preterm blood volume without deferring clamping.[32] Yet, a preterm lamb model demonstrated that during cord milking there was a transient increase of carotid blood flow and pressure.[33] A recent trial comparing deferred cord clamping with cord milking was stopped early in the subgroup of extremely preterm infants (23–27 weeks), as the incidence of severe intraventricular haemorrhage was higher in the cord milking group.[34] Hence, the effect of cord milking in different populations needs further elucidation.

### Current guidelines for cord management at birth and previous reviews of aggregate data

Current uncertainties in optimal cord management strategies are reflected in varying guidelines. The WHO recommends late cord clamping[35] unless resuscitation is required, the National Institute for Health and Care Excellence (NICE) recommends waiting for 30 s to 3 min if mother and baby are stable,[36] and the International Liaison Committee on Resuscitation Council (ILCOR) recommends a delay in cord clamping of at least 1 min. If the baby is assessed as requiring resuscitation (which is the case in many preterm infants),[37] WHO recommends immediate clamping,[38] NICE recommends considering cord milking before clamping and ILCOR concludes that there is insufficient evidence to make any recommendations.[37]

A 2012 Cochrane review of timing of cord clamping for preterm births[39] included 15 trials, with 738 infants, of which one trial (with 40 infants) compared cord milking with immediate cord clamping.[32] There was heterogeneity in the timing of cord clamping and gestational age at recruitment, and data were insufficient for reliable conclusions about any of the primary outcomes of the review. A systematic review and meta-analysis published in 2018 (including 18 trials with 2834 infants) compared the effect of deferred (≥30 s) versus early (<30 s) clamping in preterm infants, and found a reduction in the primary outcome of hospital mortality by 32% (risk ratio=0.68, 95% CI 0.52 to 0.90).[16] There was heterogeneity in the definition of 'early cord clamping' ranging from less than 5 s to 25 s, and 'late cord clamping', ranging from 30 s to 180 s. Recruitment age varied from 22 weeks to 36 weeks' gestational age. Most analyses of infant and maternal morbidity were substantially underpowered.[16] The review concluded that while there is high quality evidence that deferred cord clamping improves outcomes, individual participant data (IPD) analyses are urgently needed to

further understand the benefits and potential harms of different cord management strategies, and to understand whether differential treatment options are advantageous for key subgroups of infants.[16]

This ongoing uncertainty about the optimal cord management strategy, and differential cord management strategies for key subgroups of infants (eg, for those for which resuscitation and/or stabilisation is deemed necessary, or extremely preterm infants) has led to 117 planned, ongoing or published trials (in more than 15 000 preterm babies) that are comparing a range of cord management strategies. *IPD meta-analysis* is the gold standard for combining such trial data. IPD provides larger statistical power for estimation of treatment effects of rarer secondary endpoints and enables reliable subgroup analyses to examine hypotheses about differences in treatment effect, exploring interactions between treatment-level and participant-level characteristics.[40] *Network meta-analysis (NMA)* facilitates data synthesis when there is a range of interventions available and permits indirect comparisons across all interventions by inferring the relative effectiveness of two competing treatments through a common comparator.[41 42] NMA produces relative effect estimates for each intervention compared with every other intervention in the network. These effect sizes can be used to obtain rankings of the effectiveness of the interventions.[43] Using IPD in NMA (as opposed to aggregate data) can improve precision, increase information and reduce bias.[44]

## Objectives

The aims of this study are:

1. To evaluate the effectiveness of cord management strategies for preterm infants on neonatal mortality and morbidity, and to evaluate patient-level modifiers of treatment effect.
2. To evaluate, compare and rank the effectiveness of different cord management strategies for preterm infants on mortality and the key secondary outcomes intraventricular haemorrhage (any grade) and infant blood transfusions (any).

## METHODS AND ANALYSIS

We will conduct a systematic review of randomised trials with IPD using pairwise meta-analysis and NMA, and a nested prospective meta-analysis. The lead investigator for all potentially eligible studies will be contacted and invited to collaborate and join the IPD *Cord Management* at *P*reterm birth (iCOMP) Collaboration. Eligible trials identified up to February 2019 are listed in online supplementary file 1. The Collaboration will undertake this project according to the methods recommended by the Cochrane IPD, Multiple Interventions, and Prospective Meta-Analysis Methods Groups.[40 45 46] Reporting guidelines for NMA protocols by Chaimani *et al*[47] and Preferred Reporting Items for Systematic Reviews and Meta-Analysis (PRISMA) extension for protocols[48 49] have been followed

for reporting (PRISMA-Protocols (PRISMA-P) checklist provided in online supplementary file 2).

## Eligibility criteria
### Types of studies

Studies will be included if they are randomised trials. Cluster-randomised and quasi-random studies will be excluded. Studies must compare at least two of the interventions of interest (defined below).

### Trial participants

Participants will be women giving birth preterm (before 37 completed weeks' gestation) and/or their babies. Individually randomised studies will be eligible for inclusion if the unit of randomisation was either the woman, or the baby. Women and babies will be included regardless of whether mode of delivery was vaginal or caesarean, and whether the birth was singleton or multiple. Correlations between multiples will be accounted for in the analyses. Babies will be included regardless of whether or not they received immediate resuscitation at birth.

### Types of interventions and comparators in pairwise meta-analysis

For the pairwise meta-analysis we will include all trials that compare an intervention to enhance umbilical blood flow or allow more time for physiological transition to the comparator immediate cord clamping. This includes interventions assessing cord management strategies for timing of cord clamping, and other strategies such as cord milking. Trials will be included regardless of whether initial neonatal care is provided with the umbilical cord intact, or not. Different strategies (ie, deferred cord clamping and cord milking) will be analysed in separate subgroups to assess comparability between the groups by assessing subgroup effects and heterogeneity. They will then be collapsed into one 'cord management intervention' group if they are deemed comparable based on the previous subgroup assessments. If they are deemed non-comparable they will be analysed and interpreted separately.

### Types of interventions and comparators in NMA

For the NMA we will include, as interventions of interest, strategies for timing of cord clamping, and other cord management strategies to influence umbilical flow and placental transfusion.

Thus, interventions of interest include:
► Immediate cord clamping without milking (≤15 s or trialist defined).
► Short deferral of cord clamping (>15 s to <45 s) without milking.
► Medium deferral of cord clamping (≥45 to <90 s) without milking.
► Long deferral of cord clamping (≥90 s) without milking.
► Cord milking or stripping before immediate cord clamping (intact cord milking).
► Cord milking or stripping before deferred cord clamping (intact cord milking).

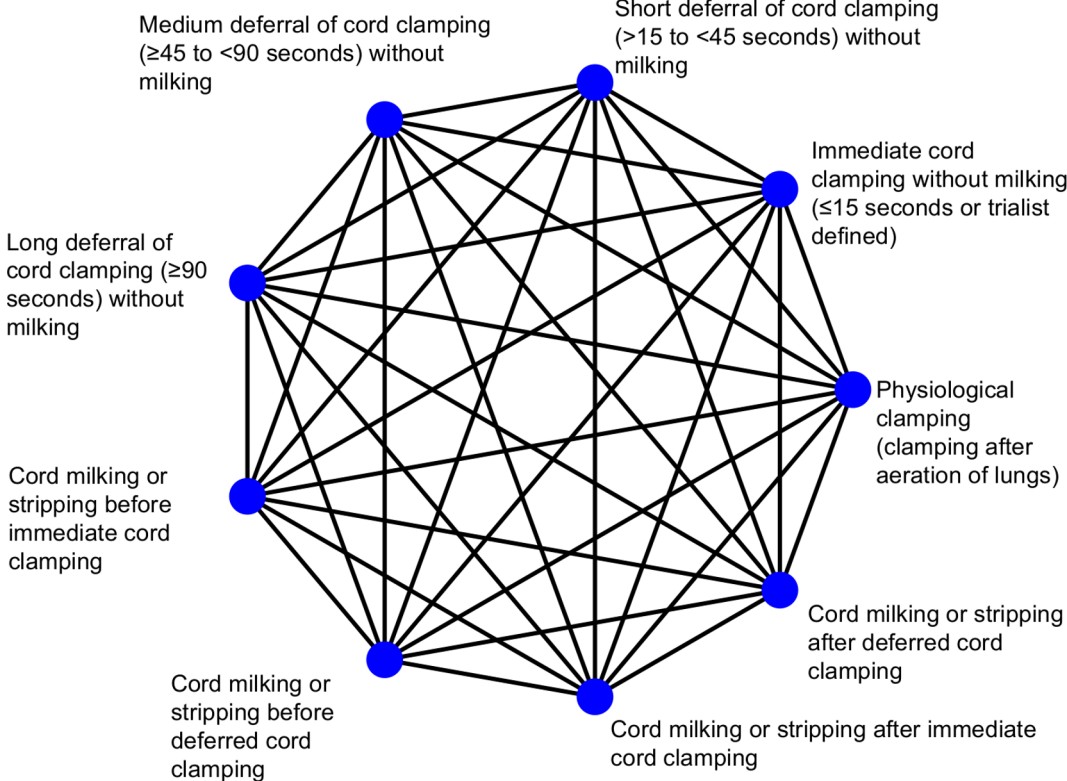

**Figure 1** Network of possible comparisons between cord management interventions.

► Cord milking or stripping after immediate cord clamping (cut cord milking).
► Cord milking or stripping after deferred cord clamping (cut cord milking).
► Physiological clamping (clamping after aeration of lungs).

If we identify other interventions not listed above we will include them if they are addressing cord management or related strategies to influence umbilical flow and placental transfusion. Again, trials will be included regardless of whether initial neonatal care is provided with the umbilical cord intact, or not. Studies evaluating collection and storage of residual placental blood that is then used for transfusion after birth will be excluded. All possible comparisons between eligible interventions are displayed in figure 1. For interpretation purposes, immediate cord clamping will act as the basis comparison/parameter.

Nodes that specify different timings of cord clamping were defined according to what timing is classified as immediate clamping, short deferral, medium deferral or long deferral according to the literature to date (as shown in online supplementary file 1), and after discussion with clinicians. Different timings are commonly compared in head-to-head comparisons, hence, their classification as different intervention nodes. Similarly, nodes that specify cord milking were classified after a review of current milking techniques described in the literature and after discussion with clinicians. If insufficient data are available, categories will be collapsed where possible. For instance, milking before and after immediate cord clamping could be collapsed into one single immediate cord milking category, or medium and long delay could be collapsed into a medium to long delay category. We consider the interventions of interest to be jointly randomisable (ie, each participant could, in principle, be randomised to any one of the interventions of interest).

### Types of outcome measures
Trials must report at least one of the clinical outcomes included in this review, as specified in the "measures" section below, to be included.

### Eligibility for nested prospective meta-analysis
Studies are only included in the nested prospective meta-analysis if the investigator/s were blind to outcome data by intervention group at the time the main components of the protocol (ie, objectives, aims and hypotheses, eligibility criteria, subgroup and sensitivity analyses, and main outcomes) were initially agreed in January 2015.

### Information sources and search strategy
The search strategy to identify potentially eligible studies includes a search of the Cochrane Collaboration Pregnancy and Childbirth Review Group's Trial Register. This register contains trials identified from: monthly searches of the Cochrane Central Register of Controlled Trials (CENTRAL) and CINAHL (EBSCO); weekly searches of Medline (Ovid) and Embase (Ovid); hand searches of specialty journals and major conferences proceedings; and current awareness alerts from further journals and BioMed Central. Further details can be found elsewhere.[50] We will identify ongoing trials that may be

eligible by searching for published protocols in Medline and Embase, searching online registries of clinical trials, and personal contacts (for example, by asking collaborators to notify any unregistered studies they are aware of). The Chief Investigators of eligible trials will be invited to join the iCOMP Collaboration. They will also be asked if they know of any further planned, ongoing or completed studies. Databases will be searched from their inception. Preliminary searches using this search strategy have already been completed up to 13 February 2019, but the search will be updated regularly to include additional trials. The search strategy is outlined in more detail in online supplementary file 3.

### Selection of studies for inclusion in the review

Two members of the iCOMP Secretariat (see project management section below) will independently assess all the potentially eligible studies identified for inclusion. Disagreements will be resolved by discussion or, if required, by consulting a third member of the iCOMP Secretariat. Studies that are not willing or able to provide IPD will be synthesised where possible using aggregate data.

### Data collection, management and confidentiality
#### Data receipt

Each participating trial will be asked to provide deidentified, individual participant-level data. Clear instructions will be provided on which data are needed and the secure data transfer process. The preferred data format and coding for each variable will be supplied to the investigators, but data in any format that is most convenient will be accepted and recoded if necessary. Data management will comply with the University of Sydney Data Management Policy 2014. Depending on trialists' preference, data transfer will either take place via secure data transfer platforms, or will be shared via institutional secure email using password-protected zip files. Data for this project will be stored in perpetuity in a password-protected folder within the NHMRC (National Health and Medical Research Council) Clinical Trials Centre's network. Only authorised project team members working within the NHMRC Clinical Trials Centre will have access to these data.

#### Data processing

*Data checking:* For each trial, range and internal consistency of all variables will be checked. Intervention details and missing data will be checked against any protocols, published reports and data collection sheets. Integrity of the randomisation process will be assessed by examining the chronological randomisation sequence and the balance of participant characteristics across intervention groups. Any inconsistencies or missing data will be discussed with the trialists and resolved by consensus. Each included study will be analysed separately and the results sent to the trial investigators for verification prior to inclusion in the iCOMP database. All trial-specific

outcomes generated from the IPD will be cross-checked against published information via a series of crosstabs.

*Data recoding:* Outcome data may have been collected in different formats across trials. Therefore, the deidentified data from each of the trials will be extracted and reformatted into a commonly coded data set.

*Data transformation and collating:* Once the data from each of the trials are finalised, they will be combined into a single data set, but a trial identifier code for each participant will be retained. New variables will be generated from the combined data set as required to address the hypotheses to be tested.

### Risk of bias assessment and certainty of evidence appraisal

Eligible studies will be assessed for risk of bias using the criteria described in the Cochrane Handbook:[51] random sequence generation; allocation concealment; blinding of participants and personnel; blinding of outcome assessment; incomplete outcome data; selective reporting; and other bias. Uncertainties will be resolved where possible by contacting study authors. Certainty of evidence will be appraised using the Grading of Recommendations Assessment, Development and Evaluation (GRADE) approach[52] for the pairwise comparisons, and the rating approach suggested by Salanti *et al* that is implemented in the Confidence in Network Meta-Analysis (CINeMA) application for the NMA.[53]

### Outcomes measures for pairwise meta-analysis

All outcome measures for the pairwise meta-analysis are listed in table 1. The primary outcome will be death of the baby prior to hospital discharge. As outcomes for babies born very preterm (before 32 weeks' gestation) are different to those born moderately preterm (32–37 weeks), separate analyses will be conducted for these two groups of infants for the secondary outcomes. Where possible, definitions will be standardised, otherwise outcomes will be used as defined in individual trials. Secondary outcomes will include measures of neonatal and maternal morbidity, and health service use.

### Covariates and subgroups for pairwise meta-analysis

Subgroup analyses will be conducted for the primary outcome of death (prior to hospital discharge) and two key secondary outcomes (intraventricular haemorrhage any grade and any infant blood transfusion). All included covariates and subgroups are listed in table 1. The comparative effects of alternative cord management strategies may vary depending on key infant risk factors, and/or on the level and type of neonatal care available at the hospital of birth. Thus, there will be subgroup analyses based on participant-level characteristics and based on hospital-level characteristics. If data are insufficient for the prespecified subgroup analyses, categories will be collapsed.

### Data analysis for pairwise meta-analysis

The full Statistical Analysis Plan will be agreed on by the iCOMP Collaboration before any analyses are undertaken. Analyses will include all randomised participants

| Table 1 | Measures for individual participant data pairwise meta-analysis |
|---|---|
| **Outcomes** | |
| **For all infants** | |
| Primary outcome | Death prior to hospital discharge |
| **For infants born before 32 weeks' gestation** | |
| Key secondary outcomes | ▶ Death (at any time during follow-up)<br>▶ Severe intraventricular haemorrhage on cranial ultrasound (grade 3–4)<br>▶ All grades of intraventricular haemorrhage on cranial ultrasound<br>▶ Necrotising enterocolitis ≥grade 2 (or trialist definition)<br>▶ Late-onset sepsis (where possible defined as clinical sepsis >72 hours after birth)<br>▶ Patent ductus arteriosus requiring treatment (medical and/or surgical)<br>▶ Chronic lung disease (at 36 weeks' postmenstrual age or trialist defined)<br>▶ Blood transfusion (yes/no) |
| Other secondary outcomes | ▶ Death (within 7 days)<br>▶ Other forms of white matter brain injury (eg, periventricular leukomalacia, porencephaly)<br>▶ Respiratory support (mechanical ventilation, CPAP, low nasal flow oxygen)<br>▶ Duration of respiratory support<br>▶ Supplemental oxygen at 36 weeks<br>▶ Retinopathy of prematurity requiring treatment (medical and/or surgical)<br>▶ Drug treatment for hypotension (yes/no)<br>▶ Blood transfusion (number, volume)<br>▶ Hypothermia on admission to neonatal unit<br>▶ Haemoglobin<br>▶ Haematocrit<br>▶ Polycythaemia<br>▶ Jaundice requiring treatment<br>▶ Birth weight<br>▶ Length of stay in NICU/SCU<br>▶ Length of stay in hospital<br>▶ Apgar scores at 1 min and 5 min<br>▶ Long-term developmental disability (assessed using the Bayley III, and/or other tools):<br> – Cerebral palsy<br> – Neurodevelopmental disability<br> – Score on cognitive scale<br> – Score on language scale<br> – Score on social/emotional scale<br> – Score on motor scale<br> – Score on behaviour scale<br> – Deafness<br> – Blindness |
| **For infants born at or after 32 weeks' gestation** | |
| Key secondary outcomes | ▶ Death at any time (during follow-up)<br>▶ Admission to NICU<br>▶ Blood transfusion (any, number, volume) |
| Other secondary outcomes | ▶ Death (within 7 days)<br>▶ Haemoglobin<br>▶ Haematocrit<br>▶ Jaundice requiring treatment<br>▶ Length of stay in NICU/SCU<br>▶ Length of stay in hospital<br>▶ Duration of respiratory support (mechanical ventilation, CPAP, low flow nasal oxygen)<br>▶ Chronic lung disease<br>▶ Late-onset sepsis (>72 hours after birth)<br>▶ Patent ductus arteriosus requiring treatment (medical and/or surgical)<br>▶ Drug treatment for hypotension<br>▶ Hypothermia on admission to neonatal unit or postnatal ward<br>▶ Apgar score at 1 min and 5 min<br>▶ Long-term developmental disability (assessed using the Bayley III, and/or other tools):<br> – Cerebral palsy<br> – Neurodevelopmental disability<br> – Score on cognitive scale<br> – Score on language scale<br> – Score on social/emotional scale<br> – Score on motor scale<br> – Score on behaviour scale<br> – Deafness<br> – Blindness |

**Table 1** Continued

| Outcomes | |
|---|---|
| **For all women** | |
| Secondary outcomes | ► Maternal death<br>► Postpartum haemorrhage<br>► Postpartum sepsis requiring treatment<br>► Manual removal of placenta<br>► Retained placenta<br>► Not breastfeeding when baby discharged from hospital<br>► Postnatal depression<br>► Blood transfusion |
| **Covariates/subgroups** | |
| *Participant-level characteristics* | ► Gestation at birth<br>► Type of pregnancy: singleton; multiple<br>► Maternal age<br>► Mode of birth: caesarean before onset of labour; caesarean after onset of labour; vaginal<br>► Onset of labour: spontaneous onset or spontaneous prelabour ruptured membranes; not spontaneous onset or spontaneous prelabour ruptured membranes; not known whether spontaneous onset of labour or spontaneous prelabour ruptured membranes<br>► Type of breathing onset: spontaneous breathing onset; supported lung aeration (ventilation); unknown<br>► Time of breathing onset relative to cord clamping: before cord clamping/milking; after cord clamping/milking; unknown<br>► Sex (male, female, uncertain/other)<br>► Ethnicity (trialist defined)<br>► Small for gestational age (trialist defined): yes/no<br>► Maternal antenatal/intrapartum/postpartum sepsis requiring treatment (trialist defined): yes/no<br>► Assessed as needing resuscitation and/or stabilisation (yes/no)<br>► Type of uterotonic drug (if any) |
| *Hospital/trial-level characteristics* | ► Highest level of neonatal unit available at site: NICU, neonatal unit (some capacity to provide ventilation), special care baby unit (no ventilation available), no neonatal unit or special care baby unit<br>► Planned timing of uterotonic drug: before cord clamping; after/at cord clamping; timing mixed or not known<br>► Planned position of the baby relative to the placenta while cord intact: level with placenta (between level of woman's bed and her abdomen/anterior thigh); more than 20 cm below level of placenta; position mixed or not known<br>► Need for immediate resuscitation at birth: infants requiring immediate resuscitation at birth excluded; infants requiring immediate resuscitation at birth included; unclear whether infants requiring immediate resuscitation at birth included or excluded<br>► Type of consent: waiver of consent; deferred consent; informed consent or assent; type of consent unclear<br>► Study year |

CPAP, continuous positive airway pressure; NICU, neonatal intensive care unit; SCU, special care unit.

for which data are available, and the primary analyses will be based on intention-to-treat. Analyses will be conducted using the open-source software R.[54]

For each outcome, a one-stage approach to analysis will be employed to include IPD from all eligible trials in a multilevel random-effects or mixed-effects regression model. Aggregate data will be included where IPD are unavailable.[55] Relative heterogeneity of treatment effects across trials will be estimated using $I^2$, with further inclusion in secondary models of participant-level and trial-level covariates to explain the sources of heterogeneity. Prediction intervals will be estimated to ascertain absolute heterogeneity. Forest plots will be presented by trial for the primary outcome, and for any secondary outcomes where there is evidence of heterogeneity across trials.

We will use a generalised linear modelling framework, with the choice of outcome distribution and link function dependent on outcome type. For example, binomial with log link will be used to estimate risk ratios for binary outcomes, and Gaussian with identity link for mean differences, with log-transformation of the data if appropriate. We will follow a similar approach for secondary outcomes. For estimation of subgroup effects, we will present forest plots of pooled treatment effects according to prespecified subgroup variables, and estimate effects by including appropriate interaction terms between a subgroup variable and treatment arm in the regression models. The results of all comparative analyses will be presented using appropriate estimates of treatment effect along with 95% CIs and two-sided p values.

**Outcome measures for NMA**

The primary outcome for the NMA will be death of the baby during the initial hospital stay. If data availability permits, IVH (any grade) and blood transfusion (any) will be analysed as two key secondary outcomes.

## Covariates and subgroups for NMA

Subgroup analyses will be conducted for the primary outcome (death before discharge) and the two key secondary outcomes (IVH any grade, blood transfusion). Gestational week at birth and highest level of available care will be considered as effect modifiers to improve consistency of the NMA model. There will be subgroup analyses assessing treatment effect by week of gestational age, and by comparing babies assessed as in need of immediate resuscitation versus not in need of immediate resuscitation.

## Assessment of the transitivity assumption for NMA

Transitivity in the network will be assured by only including interventions that are regarded as jointly randomisable and by limiting our sample to preterm infants. Gestational age at birth, hospital setting (highest level of available neonatal care), as well as study year may act as effect modifiers and could influence the transitivity of the network. We will therefore investigate whether these variables are distributed evenly across comparisons. If we find any of those variables to be unevenly distributed, they will be included in the network as covariates to investigate their influence on the network and on possible inconsistency.

## Data analysis for NMA

As for the pairwise meta-analysis, all analyses will be specified a priori in a full Statistical Analysis Plan, all randomised participants for which data are available will be included, and the primary analyses will be intention-to-treat. Again, aggregate data will be included where IPD are unavailable.

We will calculate a two-step random-effects contrast-based network meta-regression to compare and rank all available interventions for the primary outcome death (during initial hospital stay) and for the two key secondary outcomes—IVH (any grade) and blood transfusion (any). Summary risk ratios with CIs and prediction intervals will be presented for each pairwise comparison in a league table. We will estimate the ranking probability of each intervention being at each rank, and we will use surface under the cumulative ranking curve and mean ranks to obtain a treatment hierarchy. A frequentist approach to analysis will be used. Should models not converge, a Bayesian approach will be used instead, setting a weakly informative prior $d \sim N(0, 5)$. Correlations induced by multiarm studies will be accounted for using multivariate distributions.

As a second step, interactions between key covariates and effect estimates will be tested, assuming a common interaction across comparisons. If there are statistically significant interactions between covariates and treatment effects, we will provide probability rankings of intervention effects by subgroup for these covariates. A single heterogeneity parameter will be assumed for each network. Statistical heterogeneity will be assessed using the $I^2$ statistic.

## Assessment of inconsistency for NMA

Global consistency will be assessed using the Q statistics for inconsistency and the design-by-treatment interaction model. Local consistency will be assessed using the loop-specific approach and the node-splitting approach to explore sources of inconsistency. Since tests of inconsistency are known to have low power, results will be interpreted with caution, and potential known sources for inconsistency will be explored even if there is no statistical evidence of inconsistency. Any detected inconsistency will be explored by including covariates specified above (gestational age at birth, hospital setting, as well as study year) into the model, and by excluding potential outlier studies in sensitivity analyses. In case of a judgement of excessive heterogeneity or inconsistency we would still report the resulting parameters, but would interpret the results as not reliable.

## Assessment of compliance with the allocated intervention

Compliance with the interventions will be described for each trial. For studies of early versus deferred cord clamping this will be based on (1) The time to cord clamping in each allocated group. (2) The difference in time between early and deferred clamping. For studies comparing cord milking with no milking, this will be based on (1) Time to cord clamping in the allocated groups. (2) Reported compliance with cord milking in both groups.

## Assessment of selection bias

We will perform a nested prospective meta-analysis as a sensitivity analysis, to detect potential differences between prospectively and retrospectively included studies that may point to selection or publication bias. We expect to also be able to include some unreported outcomes sourced from the IPD provided by the included studies, which may alleviate selective outcome reporting bias.[40] Additionally, comparison-adjusted and contour-enhanced funnel plots[56] will be used to examine whether there are differences in results between more and less precise studies.

## Adjustments for multiplicity

There is only one primary outcome, and few key secondary outcomes for this study. For other secondary outcomes, no formal adjustments for multiplicity (ie, the accumulation of type 1 error and thus higher likelihood of chance findings when assessing multiple outcomes) are planned. Instead, we will be taking the following approach outlined by Schulz and Grimes:[57] as secondary outcomes examined in this study are interrelated, we will interpret the pattern of results, examining consistency of results across related outcomes, instead of focusing on any single, statistically significant result. All secondary outcomes will be reported. Subgroup analyses will be performed by testing for interactions and findings will be reported as exploratory.[58]

## Planned sensitivity analyses

To assess whether results are robust to trial characteristics and methods of analysis, the following sensitivity analyses will be conducted for the primary outcome, if data are sufficient:

► Excluding trials with high risk of bias for sequence generation and/or concealment of allocation and/or loss to follow-up for pairwise meta-analysis and NMA.
► Excluding trials with a significant conflict of interest (eg, trials funded by pharmaceutical companies).
► For trials comparing early cord clamping with deferred clamping, analysis of outcomes weighted by degree of separation in mean actual timing of cord clamping between intervention and control groups for pairwise meta-analysis.
► Analysis of outcomes weighted by degree of separation in haemoglobin (at 24 hours) achieved between intervention and control groups for pairwise meta-analysis (as a surrogate for net placental transfusion).
► For trials with deferred cord clamping, an additional dose-response analysis assessing intended time of cord clamping deferral as a continuous variable will be performed.
► Exploratory analysis based on actual, rather than intended, timing of cord clamping for individual participants for pairwise meta-analysis and NMA.
► The impact of missing data on the effects of the included interventions for the primary outcome may be explored (if appropriate).

## Project management

The iCOMP Collaboration will invite membership from representatives of each of the included trials contributing IPD, have a Secretariat, and invite methodological and clinical experts who will form an Advisory Group. The Secretariat will be responsible for data collection, management and analysis, and for communication within the Collaboration.

## Public and patient involvement

Two consumer representatives have been invited to join the iCOMP collaboration, comment on this protocol and be involved in the interpretation of results.

## Ethical issues

For each included trial, ethics approval has been previously granted by their respective Human Research Ethics Committees (or equivalent), and informed consent has been obtained from all participants. The Chief Investigators of the included trials remain the custodians of their own trial's data. IPD from the included trials will be deidentified before sharing with the iCOMP Collaboration.

## Publication policy

The key methods for this meta-analysis protocol were agreed by the iCOMP Collaborators in January 2015, before unblinding of any outcome data from the studies included in the nested prospective meta-analysis. This manuscript was discussed at the iCOMP Collaborators' meeting held at the Paediatric Academic Societies meeting in San Diego in April 2015. At this meeting it was agreed the protocol should be expanded to include a retrospective systematic review and IPD and NMA with a nested prospective meta-analysis. The protocol was then revised, based on further discussion, and circulated to members

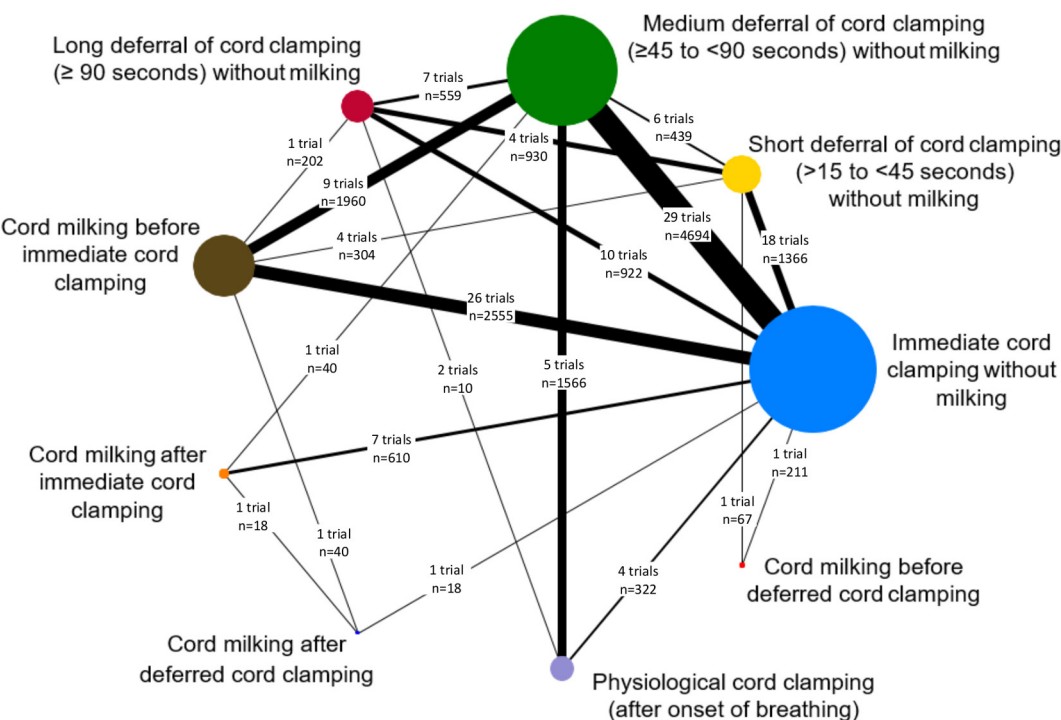

**Figure 2** Illustration of network of currently available trials comparing different timings of cord clamping.

of the iCOMP Collaboration for further comment and agreement prior to manuscript submission.

Participating trialists in the prospective meta-analysis, when reporting results from their own trials, will endeavour to include a statement that their trial is part of this prospective meta-analysis in any published manuscripts or conference abstracts. Any reports of the results of this meta-analysis will be published either in the name of the collaborative group, or by representatives of the collaborative group on behalf of the iCOMP Collaboration, as agreed by members of the collaborative group. Draft reports will be circulated to the collaborative group for comment and approval before submission for publication.

## DISCUSSION

There is an urgency to conduct this systematic review and pairwise IPD and NMA so we can make sense of the numerous trials currently being undertaken, inform clinical practice and identify the most promising interventions for further evaluation.

This meta-analysis offers an opportunity to reliably test important hypotheses that cannot be resolved by any of the individual trials, either alone or in simple combination. Coordinating international efforts in this way will help achieve consensus on the most important substantive clinical outcomes to assess in any future trials as needed. Unequivocal synthesised results, together with the identification of key determinants (eg, effect modifiers), will be critical for translating evidence from this meta-analysis directly into practice. Figure 2 shows the network of comparisons available from the trials identified to date. We plan to complete study identification and IPD collection by early 2020, then conduct the analyses and disseminate results by mid-2021. Trials that are ongoing and therefore unable to provide data by March 2020 will remain members of the iCOMP collaboration. Their data will be included in future updates of iCOMP.

This study is only possible because trialists around the world have agreed to collaborate to share the IPD from their cord management trials. This collaborative approach will enable us to move beyond the traditional 'one-size-fits-all' and towards precision medicine, to find the optimal intervention from a range of treatment options for each individual woman and her baby, based on their individual characteristics and risk factors.

## Author affiliations
[1]NHMRC Clinical Trials Centre, The University of Sydney, Sydney, New South Wales, Australia
[2]Nottingham Clinical Trials Unit, University of Nottingham, Nottingham, UK
[3]Neonatal Research Institute, Sharp Mary Birch Hospital for Women & Newborns, San Diego, California, USA
[4]Department of Obstetrics and Gynecology, Clinic University Hospital Virgen de la Arrixaca, Murcia, Spain
[5]Department of Paediatrics and Child Health, Cork University Maternity Hospital, Cork, Ireland
[6]Academic Department of Paediatrics, Brighton and Sussex University Hospitals, Brighton, UK
[7]Department of Pediatrics and Medicine, University of Virginia, Charlottesville, Virginia, USA
[8]College of Nursing, University of Rhode Island, Kingston, Rhode Island, USA
[9]Department of Pediatrics, St Louis University School of Medicine, St Louis, Missouri, USA
[10]Department of Clinical Sciences Lund, Pediatrics/Neonatology, Skane University Hospital, Lund University, Lund, Sweden
[11]Department of Perinatal and Neonatal Medicine, Jichi Medical University Saitama Medical Center, Saitama, Japan
[12]Newborn Unit, Department of Pediatrics, Postgraduate Institute of Medical Education and Research, Chandigarh, India
[13]Department of Neonatology, Lady Hardinge Medical College, New Delhi, India
[14]Department of Pediatrics, Dalhousie University, Halifax, Nova Scotia, Canada
[15]Julius Center for Health Sciences and Primary Care, University Medical Center Utrecht, Utrecht, The Netherlands
[16]The Ritchie Centre, Obstetrics & Gynaecology, Monash University, Clayton, Victoria, Australia
[17]Department of Neonatology, University of Sydney, Sydney, New South Wales, Australia
[18]Newborn Research Centre, The Royal Women's Hospital, Melbourne, Victoria, Australia

**Acknowledgements** The authors thank Sarah Somerset, Min Yang, Charlotte Lloyd and Virginia Portillo for previous support for the Secretariat and input into earlier protocol drafts.

**Collaborators** iCOMP Collaboration members: Secretariat: Angie Barba, Anna Lene Seidler, Ava Grace Tan-Koay, Kylie E Hunter, Lisa Askie (NHMRC Clinical Trials Centre, University of Sydney, Sydney, Australia); Alan Montgomery, Lelia Duley (Nottingham Clinical Trials Unit, University of Nottingham, Nottingham, UK). Trialists: Amir Kugelman (Bnai Zion Medical Center, Haifa, Israel); Anu George (Malankara Orthodox Syrian Church Medical College, Kerala, India); Anu Sachdeva (All India Institute of Medical Science, New Delhi, India); Anup Katheria (Sharp Mary Birch Hospital for Women & Newborns, San Diego, California, USA); Arjan Te Pas (Leiden University, Leiden, The Netherlands); Ashish K C (Uppsala University, Uppsala, Sweden); Bimlesh Kumar (Lala Lajpat Rai Memorial Medical College, Uttar Pradash, India); Carl Backes (Ohio State University Wexner Medical Center, Columbus, Ohio, USA); Catalina De Paco Matallana (Clinic University Hospital Virgen de la Arrixaca, Murcia, Spain); Chamnan Tanprasertkul (Thammasat University, Pathumthani, Thailand); Chayatat Ruangkit (Mahidol University, Samut Prakan, Thailand); Eugene Dempsey (Cork University Maternity Hospital, Cork, Ireland); G Ram Mohan, Lakhbir Dhaliwal, Venkataseshan Sundaram (Post Graduate Institute of Medical Education & Research, Chandigarh, India); Gillian Gyte (Oldfield, Poulton le Fylde, UK), Guillermo Carroli (Rosarino Center for Perinatal Studies, Rosario, Argentina); Heidi Al-Wassia (King Abdulaziz University, Jeddah, Saudi Arabia); Hytham Atia (Zagazig University, Zagazig, Egypt); Heike Rabe (Brighton and Sussex Medical School, University of Sussex, Brighton, UK); Islam Nour (Mansoura University Children's Hospital, Mansoura, Egypt); Jiangqin Liu (Tongji University School of Medicine, Shanghai, China), John Kattwinkel, Karen Fairchild (University of Virginia, Charlottesville, Virginia, USA); Judith Mercer (The University of Rhode Island, Kingston, Rhode Island, USA), Justin Josephsen (St Louis University School of Medicine, St. Louis, Missouri, USA), Kellie Murphy (Mount Sinai Hospital, Toronto, Canada); Kristy Robledo, William Tarnow-Mordi (NHMRC Clinical Trials Centre, University of Sydney, Sydney, Australia); Laura Perretta (Weill Cornell Medical College, New York City, New York, USA); Lin Ling (Suining Central Hospital, Sichuan, China); Manoj Varanattu (Jubilee Mission Medical College & Research Centre, Kerala, India); Maria Goya (Vall d'Hebron University Hospital, Barcelona, Spain); Michael Meyer (Middlemore Hospital, Auckland, New Zealand); Musa Silahli (Baskent University, Ankara, Turkey); Neelam Kler (Sir Ganga Ram Hospital, New Delhi, India); Neil Finer (University of California, San Diego, California, USA); Ola Andersson (Skane University Hospital, Lund University, Lund, Sweden); Omar Kamlin, Shiraz Badurdeen (The Royal Women's Hospital, Melbourne, Australia); Pharuhad Pongmee (Ramathibodi Hospital, Mahidol University, Bangkok, Thailand); Prisana Panichkul, Sangkae Chamnanvanakij (Phramongkutklao Hospital, Ratchathewi, Bangkok, Thailand); Ronny Knol (Erasmus MC, University Medical Centre, Rotterdam, The Netherlands); Sandeep Kadam (KEM Hospital, Pune, Maharashtra, India); Shigeharu Hosono (Jichi Medical University Saitama Medical Center, Saitama, Japan); Simone Pratesi (University of Florence, Florence, Italy); Thomas Ranjit (St John's Medical College & Hospital, Bangalore, Kamataka, India); Victor Lago Leal (University Hospital of Getafe & European University of Madrid, Madrid, Spain); Vikram Datta

(Lady Hardinge Medical College, New Delhi, India); Waldemar Carlo (University of Alabama at Birmingham, Birmingham, Alabama, USA), Walid El-Naggar (Dalhousie University, Halifax, Canada). Advisers: Graeme Polglase, Stuart Hooper (The Ritchie Centre, Obstetrics & Gynaecology, Monash University, Melbourne, Australia); John Simes (NHMRC Clinical Trials Centre, University of Sydney, Sydney, Australia); Martin Kluckow (University of Sydney, Sydney, Australia); Peter Davis (The Royal Women's Hospital, Melbourne, Australia); Thomas P.A. Debray (Julius Center for Health Sciences and Primary Care, University Medical Center Utrecht, Utrecht, the Netherlands)

**Contributors** LD and LA conceived the idea. ALS, LA and LD drafted the protocol. All remaining authors (ACK, CDPM, ED, HR, JK, JM, JJ, KF, OA, SH, VS, VD, WE-N, WT-M, TD, SBH, MK, GP, PGD, AM, KEH, AB and JS) critically revised all drafts of the manuscript for intellectual content, and agreed and approved the final manuscript. ALS is the guarantor of the review.

**Funding** Support for developing the protocol and establishing the collaborative group was received from the UK National Institute of Health Research with a grant entitled "The Preterm Birth Programme" (number RPPG060910107). This grant presents independent research commissioned by the National Institute for Health Research (NIHR) under its Programme Grants for Applied Research funding scheme (RP-PG0609-10107). The views expressed are those of the authors and not necessarily those of the NHS, the NIHR or the Department of Health. Funding for individual trials remains the responsibility of the trialists themselves. Funding to undertake data collection and data analysis for the iCOMP Collaboration has been provided by the Australian National Health and Medical Research Council via a Project Grant (APP1163585).

**Competing interests** LD, ACK, CDPM, ED, HR, JK, JM, JJ, KF, OA, SH, VS, VD, WE-N and WT-M are Chief Investigators for potentially eligible trials.

**Patient consent for publication** Not required.

**Provenance and peer review** Not commissioned; externally peer reviewed.

**ORCID iDs**
Anna Lene Seidler http://orcid.org/0000-0002-0027-1623
Venkateseshan Sundaram http://orcid.org/0000-0002-3135-8115
Peter G Davis http://orcid.org/0000-0001-6742-7314
Kylie E Hunter http://orcid.org/0000-0002-2796-9220

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
