## [Reviewer comments · BMJ Open]

ARTICLE DETAILS

TITLE (PROVISIONAL)	Systematic review and network meta-analysis with individual participant data on Cord Management at Preterm Birth (iCOMP): study protocol
AUTHORS	Seidler, Anna Lene; Duley, Lelia; Katheria, Anup; De Paco Matallana, Catalina; Dempsey, E; Rabe, Heike; Kattwinkel, John; Mercer, Judith; Josephsen, Justin; Fairchild, Karen; Andersson, Ola; Hosono, Shigeharu; Sundaram, Venkateshan; Datta, Vikram; El-Naggar, Walid; Tarnow-Mordi, William; Debray, Thomas; Hooper, Stuart B.; Kluckow, Martin; Polglase, Graeme; Davis, Peter; Montgomery, Alan; Hunter, Kylie; Barba, Angie; Simes, John; Askie, Lisa

VERSION 1 – REVIEW

REVIEWER	simone pratesi Azienda Ospedaliera Universitaria Careggi, Firenze, Italy
REVIEW RETURNED	17-Oct-2019

GENERAL COMMENTS	The authors propose a very useful project, that is a very comprehensive review of interventions for umbilical cord management in preterm infants. In fact, there are multiple competing cord management strategies, such as clamping the cord at different times or milking the cord, and considerations of the infant's respiratory status, and it is currently unknown which strategy yields the best balance of benefits and harms. Thus, the findings of such a review will be really highly relevant to clinicians and guideline developers. Authors will conduct a systematic search for all planned, ongoing and completed randomised controlled trials comparing alternative cord management strategies at preterm birth (before 37 weeks' gestation). First, deferred clamping and cord milking will be compared with immediate clamping in pairwise individual participant data (IPD) meta-analyses. Second, all identified cord management strategies will be compared and ranked in an IPD Network meta-analysis for the primary outcome and the key secondary outcomes intraventricular haemorrhage (any grade) and infant blood transfusions (any). The study protocol is very interesting and methodologically very well described. I suggest for it publication. I have only a comment about the supplementary File 1: in the table are listed all trials the authors identified up to February 2019 as being eligible for inclusion in iCOMP. In that list is not included an ongoing RCT being performed in Italy ("Delivery Room Assistance With the Placental Circulation Intact (PCI-T)", ClinicalTrials.gov Identifier: NCT02671305) that could be eligible for their review. Can the authors comment about that?
--

REVIEWER	Michael P Meyer Neonatal Unit, KidzFirst Middlemore Hospital Auckland New Zealand Paediatrics Child and Youth Health, University of Auckland, Auckland, New Zealand
REVIEW RETURNED	23-Oct-2019

GENERAL COMMENTS	This protocol describes a study to attempt to answer an important and topical issue in perinatal medicine, which is how to choose the most appropriate cord management strategy for the individual preterm infant. The many different strategies and clinical scenarios lend themselves to individual participant data meta analysis and network meta analysis. The large number of collaborators and inclusion of studies from different types of hospitals will help ensure the results are generalizable. Specific comments:  1. Some of the identified studies are on going. Studies identified up to Feb 2019 are included in the file but more searches are planned. How will the cut off date be determined? It may be useful to include some of the following outcomes depending on how far data collection has progressed. 2. For preterm infants <32 weeks blood transfusion is not a key secondary outcome, whilst it is for infants 32 weeks and over. Transfusion associated morbidity is likely to be greater in the more preterm group. 3. Jaundice and treatment are not included as secondary outcomes in the 32 week and over group, whilst this could still be clinically important. 4. The type of anaesthetic eg spinal or general anaesthetic could be important in determining how soon the baby breathes and how long the deferred clamping continues. 5. Data collection includes singleton/multiple. It would be worth including monochorionic/ dichorionic as this another controversial area. 6. The authors might consider including whether infants were intubated in delivery room as part of the early care data.
--

REVIEWER	Stefano Ghirardello Fondazione IRCCS Ca' Granda Ospedale Maggiore Policlinico, Milan, Italy
REVIEW RETURNED	15-Nov-2019

GENERAL COMMENTS	I read the study protocol with great interest, and I believe it will provide important information to improve our knowledge of umbilical cord management and placental transfusion strategies in preterm deliveries. I have only two minor issues to address: I wonder why the published pilot study by Pratesi et al. with 40 newborns less than 30 weeks gestational age (Placental Circulation Intact Trial (PCI-T)—Resuscitation With the Placental Circulation Intact vs. Cord Milking for Very Preterm Infants: A Feasibility Study. Frontiers in Pediatrics 2018; doi: 10.3389/fped.2018.00364) was not included in this systematic review. The study by Pratesi is a multicenter trial still open to enrollment (Clinicaltrials.gov NCT02671305 (date of registration: 26 JAN 2016); the aim is to compare bedside assistance
--

	with intact placental circulation for 3min to cord milking in preterm newborns less than 30 weeks of gestational age. Actually 88 patients have been enrolled. The principal investigator may be contacted to share Individual participating data from patients enrolled in this trial. Data receipt: the authors should provide some more specific information on data transmission from the Chief investigators of eligible trials to the iCOMP group. The authors state that "De-identified, individual participant level data will be provided by each participating trial. These data will be backed-up and stored in a centralized secure database". In my opinion, this sentence does not provide sufficient information on data acquisition methods and security guarantees in the transmission of the data itself. Thank you for the opportunity to review this valuable protocol. Stefano Ghirardello
--	---

REVIEWER	Areti Angeliki Veroniki Department of Primary Education, School of Education, University of Ioannina
REVIEW RETURNED	22-Dec-2019

GENERAL COMMENTS	This is a study protocol for a systematic review and IPD-NMA to evaluate the effectiveness of cord management strategies in preterm infants. The paper is well-written, but there are some items that require noting and further clarification. I address these items below in separate bullet points.  - The authors will systematically search for planned, ongoing, and completed RCTs. Although this sounds a very nice plan, I find it hard for RCT authors to share their data when their RCT is not completed and their results published. This is because usually trialists want to publish the results themselves first. Also, older trials may not have been published due to certain problems with the RCT and trialists may be reluctant to share their data. Can the authors justify that this is a reasonable approach in their case? Also, I am unclear how the authors are going to include IPD from RCTs that are at the planning stage, since these most probably will not include any patients yet. - Abstract - "IPD will be sought for all trials": What if IPD are not obtained for all identified studies? Unfortunately, this is a very common problem (see: https://www.ncbi.nlm.nih.gov/pubmed/31153977), especially in NMA where a number of studies compare the several competing treatments included in the network. - Another important item to consider is the funding of the individual studies. How are the authors going to proceed if the studies are funded by pharmaceutical companies? What if the data are provided for a certain period of time and only through the funder's online protected platform? - Methods: An IPD-NMA should follow 2 types of PRISMA for reporting: PRISMA-IPD and PRISMA-NMA. These should certainly be followed for the reporting of the final review. The authors state that the PRISMA-IPD is followed only. However, at the protocol
---

stage the PRISMA-P is advisable to be considered for reporting, which is not stated in the text. Although the authors mention that they followed the PRISMA-IPD, the PRISMA-P checklist is reported in the supplementary. Please clarify.

- The PROSPERO registration number should be provided in the protocol and its abstract.

- The authors mention that they will search certain databases to identify relevant studies. However, in the search codes available in the supplementary, the search dates are already available. Have the searches already been completed? I am also unclear about this when having a look at the network plots. Do these include studies provided in a previous search or the current search and these will be updated?

- Methods - "Women and babies will be included regardless of whether mode of delivery was vaginal or caesarean, and whether the birth was singleton or multiple": How will the authors treat babies born from the same mother in their analysis? Will they assume that they are independent with different characteristics?

- What data will the authors request from trialists to provide them? The specific contact process needs to be described in the paper (for more details on this see: <https://www.ncbi.nlm.nih.gov/pubmed/27116943>)

- "These data will be backed-up and stored in a centralised secure database": More details are required for the data storage and protection. How long will these data be kept in this secured database? Will it be destroyed after few years? Who will have access to these data? Where will the secured database be located at?

- Details on the process of checking the available data and of producing a consistent format for all trials is missing.

- Which RoB tool will be used to assess study quality? Is it the newer ROB 2 tool? How results will be categorized? High, Low, and Unclear? In such a case, will the unclear cases be clarified with the original study authors of the IPD?

- "Analyses will be conducted using the open-source software R": Which R packages will be used to conduct the IPD meta-analysis and network meta-analysis? These should be specified.

- More details on the models to be used are required. Will the authors use a fixed or random intercept and slope? Which heterogeneity method will the authors use to estimate the between-study variance?

- "In advance of conducting the analyses, we will decide whether there are sufficient reliable data to allow meaningful analysis of any individual outcome": This is unclear - how will this be decided?

- "all analyses will be specified a-priori in a full Statistical Analysis Plan": I disagree with this. The statistical analysis plan should be defined in this protocol. All potential approaches should be clarified at this point, well before the data is obtained, and the statistical analysis plan should not be data driven.

	 - Page 13, lines 17-20: Please elaborate on what you mean by "if data permits". - "Should models not converge, a Bayesian approach will be used instead" This is unclear. What is meant by the models do not converge in a frequentist framework? - "setting a prior of no effect and a large variance" The specific priors that will be used in the Bayesian models should be provided. Also, details on the models to be applied and the software to be used should be specified at the protocol stage. - "potential known sources for inconsistency will be explored" The potential effect modifiers should be specified a priori and at the protocol stage. - "A judgement of excessive heterogeneity or inconsistency would prevent the interpretation and reporting of the network meta-analysis." Please clarify this sentence. - Citation 58 should refer to the loop-specific approach. A relevant citation for the low power of inconsistency tests is: https://www.ncbi.nlm.nih.gov/pubmed/25239546 - "We expect to also be able to include some unreported outcomes sourced from the individual participant data provided by the included studies, alleviating selective outcome reporting bias." Is there published evidence for this? Can you please provide relevant citations to back-up this statement? - "For other secondary outcomes, no formal adjustments for multiple testing are planned but instead, we will be following the approach outlined by Schulz and Grimes(60):" Please clarify what is meant by multiple testing. Also, I believe the reader will not be familiar with the Schulz and Grimes(60) approach. Please provide more details on this. - The authors should specify for which outcomes they are going to apply subgroup analyses and meta-regression analyses. - Discussion: A paragraph outlining the limitations of the present study is missing. - Patient and Public Involvement: Please clarify how the patient comments will be included in this project.
--	---

VERSION 1 – AUTHOR RESPONSE

Reviewer: 1

Reviewer Name: simone pratesi

Institution and Country: Azienda Ospedaliera Universitaria Careggi, Firenze, Italy Please state any competing interests or state 'None declared': None declared

The authors propose a very useful project, that is a very comprehensive review of interventions for umbilical cord management in preterm infants. In fact, there are multiple competing cord management

strategies, such as clamping the cord at different times or milking the cord, and considerations of the infant's respiratory status, and it is currently unknown which strategy yields the best balance of benefits and harms. Thus, the findings of such a review will be really highly relevant to clinicians and guideline developers.

Authors will conduct a systematic search for all planned, ongoing and completed randomised controlled trials comparing alternative cord management strategies at preterm birth (before 37 weeks' gestation). First, deferred clamping and cord milking will be compared with immediate clamping in pairwise individual participant data (IPD) meta-analyses. Second, all identified cord management strategies will be compared and ranked in an IPD Network meta-analysis for the primary outcome and the key secondary outcomes intraventricular haemorrhage (any grade) and infant blood transfusions (any).

The study protocol is very interesting and methodologically very well described. I suggest for it publication.

Response: Thank you very much for this positive feedback on our manuscript.

I have only a comment about the supplementary File 1: in the table are listed all trials the authors identified up to February 2019 as being eligible for inclusion in iCOMP. In that list is not included an ongoing RCT being performed in Italy ("Delivery Room Assistance With the Placental Circulation Intact (PCI-T)", <http://scanmail.trustwave.com/?c=13000&d=hd2J3gypC-RFnJlCPGSaRhU5m295fnb5kAaRAHUwfA&u=http%3a%2f%2fClinicalTrials%2egov> Identifier: NCT02671305) that could be eligible for their review. Can the authors comment about that?

Response: We have recently updated our search, and identified this trial as eligible for inclusion in iCOMP. We have since successfully invited the principal investigator to join our collaboration. This trial is now listed in Supplementary File 1, and in the iCOMP collaborator list (p.16).

Reviewer: 2

Reviewer Name: Michael P Meyer

Institution and Country: Neonatal Unit, KidzFirst Middlemore Hospital Auckland New Zealand
Paediatrics Child and Youth Health, University of Auckland, Auckland, New Zealand Please state any competing interests or state 'None declared': None declared

This protocol describes a study to attempt to answer an important and topical issue in perinatal medicine, which is how to choose the most appropriate cord management strategy for the individual preterm infant. The many different strategies and clinical scenarios lend themselves to individual participant data meta analysis and network meta analysis. The large number of collaborators and inclusion of studies from different types of hospitals will help ensure the results are generalizable.

Response: Thank you for pointing out the importance of this planned study.

Specific comments:

1. Some of the identified studies are on going. Studies identified up to Feb 2019 are included in the file but more searches are planned. How will the cut off date be determined? It may be useful to include some of the following outcomes depending on how far data collection has progressed.

Response: Thank you for this excellent point. We have set the cut-off for data provision to iCOMP for March 2020. Any trials that are unable to provide data by this date will remain members of the iCOMP collaboration. Once additional trials have been completed iCOMP will be updated to include these additional trials.

We have described this in the revised manuscript as follows:

'We plan to complete study identification and individual participant data collection by early-2020, then conduct the analyses and disseminate results by mid-2021. Trials that are ongoing and therefore unable to provide data by March 2020 will remain members of the iCOMP collaboration. Their data will be included in future updates of iCOMP.' (p.14)

2. For preterm infants <32 weeks blood transfusion is not a key secondary outcome, whilst it is for infants 32 weeks and over. Transfusion associated morbidity is likely to be greater in the more preterm group.

Response: Thank you for this excellent point, we have now included blood transfusion as a key secondary outcome in both groups.

3. Jaundice and treatment are not included as secondary outcomes in the 32 week and over group, whilst this could still be clinically important.

Response: Thank you for this excellent point, we have now included jaundice requiring treatment as a secondary outcome in both groups.

4. The type of anaesthetic eg spinal or general anaesthetic could be important in determining how soon the baby breathes and how long the deferred clamping continues.

Response: Thank you for this valuable comment. As recommended in the updated Cochrane handbook chapter on individual participant data meta-analysis, we have conducted some preliminary variable mapping to determine which variables will likely be available for combined analysis. Whilst we agree that this variable would be valuable, it is unlikely that this level of detail will be collected by the majority of the trials. After careful consideration, we have therefore decided not to include it in our meta-analysis.

5. Data collection includes singleton/multiple. It would be worth including monochorionic/dichorionic as this another controversial area.

Response: Again, preliminary variable mapping has shown that this level of detail will not be available for the majority of the trials. After careful consideration, we have therefore decided not to include it in our metaanalysis.

6. The authors might consider including whether infants were intubated in delivery room as part of the early care data.

Response: Thank you for this excellent point. We plan to collect this as part of the resuscitation data.

Reviewer: 3

Reviewer Name: Stefano Ghirardello

Institution and Country: Fondazione IRCCS Ca' Granda Ospedale Maggiore Policlinico, Milan, Italy

Please state any competing interests or state 'None declared': none declared

I read the study protocol with great interest, and I believe it will provide important information to improve our knowledge of umbilical cord management and placental transfusion strategies in preterm deliveries.

Response: Thank you very much for this positive feedback.

I have only two minor issues to address:

1. I wonder why the published pilot study by Pratesi et al. with 40 newborns less than 30 weeks gestational age (Placental Circulation Intact Trial (PCI-T)—Resuscitation With the Placental Circulation Intact vs. Cord Milking for Very Preterm Infants: A Feasibility Study. *Frontiers in Pediatrics* 2018; doi:

10.3389/fped.2018.00364) was not included in this systematic review.

The study by Pratesi is a multicenter trial still open to enrollment (Clinicaltrials.gov NCT02671305 (date of registration: 26 JAN 2016); the aim is to compare bedside assistance with intact placental circulation for 3min to cord milking in preterm newborns less than 30 weeks of gestational age. Actually 88 patients have been enrolled. The principal investigator may be contacted to share Individual participating data from patients enrolled in this trial.

Response: Thank you for pointing this out. As outlined in our response to reviewer 1, we have now included this trial in iCOMP.

2. Data receipt: the authors should provide some more specific information on data transmission from the Chief investigators of eligible trials to the iCOMP group. The authors state that "De-identified, individual participant level data will be provided by each participating trial. These data will be backedup and stored in a centralized secure database". In my opinion, this sentence does not provide sufficient information on data acquisition methods and security guarantees in the transmission of the data itself.

Response: As requested by the reviewer, we have provided additional information on data receipt and storage in the revised manuscript. The section now reads as follows:

'Each participating trial will be asked to provide de-identified, individual participant level data. Clear instructions will be provided on which data are needed and the secure data transfer process. The preferred data format and coding for each variable will be supplied to the investigators, but data in any format that is most convenient will be accepted and recoded if necessary. Data management will comply with the University of Sydney Data Management Policy 2014, and has been approved by the University of Sydney Human Research Ethics Committee (2018/886). Depending on trialists' preference, data transfer will either take place via secure data transfer platforms, or shared via institutional secure email using password protected zip-files. Data for this project will be stored in perpetuity in a password-protected folder within the NHMRC Clinical Trials Centre's network. Only authorised project team members working within the NHMRC Clinical Trials Centre will have access to these data.' (pp.8-9)

3. Thank you for the opportunity to review this valuable protocol.

Response: Thank you for your valuable comments.

Stefano Ghirardello

Reviewer: 4

Reviewer Name: Areti Angeliki Veroniki

Institution and Country: Department of Primary Education, School of Education, University of Ioannina

Please state any competing interests or state 'None declared': None declared

This is a study protocol for a systematic review and IPD-NMA to evaluate the effectiveness of cord management strategies in preterm infants. The paper is well-written, but there are some items that require noting and further clarification. I address these items below in separate bullet points.

1. The authors will systematically search for planned, ongoing, and completed RCTs. Although this sounds a very nice plan, I find it hard for RCT authors to share their data when their RCT is not completed and their results published. This is because usually trialists want to publish the results themselves first. Also, older trials may not have been published due to certain problems with the RCT and trialists may be reluctant to share their data. Can the authors justify that this is a reasonable approach in their case? Also, I am unclear how the authors are going to include IPD from RCTs that are at the planning stage, since these most probably will not include any patients yet.

Response: Thank you for this excellent point.

As described above in our response to reviewer 1 (comment 1), we have now added further information on how we will deal with trials that do not finish on time. We have set the cut-off for data provision to iCOMP for March 2020. Any trials that are unable to provide data by this date will remain members of the iCOMP collaboration. Once additional trials have been completed iCOMP will be updated to include these additional trials.

2. Abstract - "IPD will be sought for all trials": What if IPD are not obtained for all identified studies?

Unfortunately, this is a very common problem (see:

[https://scanmail.trustwave.com/?c=13000&d=hd2J3gypC-](https://scanmail.trustwave.com/?c=13000&d=hd2J3gypC-RFnJlCpGSaRhU5m295fmb5kATCCC81eQ&u=https%3a%2f%2fwww%2encbi%2enlm%2enih%2ego)

[RFnJlCpGSaRhU5m295fmb5kATCCC81eQ&u=https%3a%2f%2fwww%2encbi%2enlm%2enih%2ego](https://scanmail.trustwave.com/?c=13000&d=hd2J3gypC-RFnJlCpGSaRhU5m295fmb5kATCCC81eQ&u=https%3a%2f%2fwww%2encbi%2enlm%2enih%2ego)
v%2fpubmed%2f31153977%29 especially in NMA where a number of studies compare the several competing treatments included in the network.

Response: We agree with the reviewer that this is a common problem, and therefore will include aggregate data where IPD are unavailable. We had previously indicated this in the protocol as follows:

Strengths and limitations of this study: 'For some of the trials it will not be possible to obtain individual participant data, so published aggregate results will be used instead' (p.3)

'Studies that are not willing or able to provide IPD will be synthesised where possible using aggregate data.' (p.8)

'Aggregate data will be included where individual participant data are unavailable' (p.10)

In response to the reviewer's comment, we have now also added this important information to the revised abstract, and to the revised NMA methods section as follows:

Abstract: 'aggregate summary data will be included where IPD are unavailable' (p.2)

'Again, aggregate data will be included where IPD are unavailable.' (p. 11)

3. Another important item to consider is the funding of the individual studies. How are the authors going to proceed if the studies are funded by pharmaceutical companies? What if the data are provided for a certain period of time and only through the funder's online protected platform?

Response: The studied interventions are non-industry interventions and therefore funding by pharmaceutical companies is likely to be less of a problem than for industry interventions that are likely to generate profit. Yet, we recognise that potential bias through funding should be assessed carefully in every systematic review. We therefore added a planned sensitivity analysis excluding studies with significant conflict of interest to our revised protocol. (p.12)

If data are only provided for a certain period of time or only through an online protected platform we will extract summary data for each analysis for this study (including subgroup data), and include these data in the analysis. Yet, from our communication with trialists to date we do not expect these issues.

4. Methods: An IPD-NMA should follow 2 types of PRISMA for reporting: PRISMA-IPD and PRISMA-NMA.

These should certainly be followed for the reporting of the final review. The authors state that the PRISMA-IPD is followed only. However, at the protocol stage the PRISMA-P is advisable to be considered for reporting, which is not stated in the text. Although the authors mention that they followed the PRISMA-IPD, the PRISMA-P checklist is reported in the supplementary. Please clarify.

Response: Thank you for pointing this out. For this protocol, we followed PRISMA-P and hence included the PRISMA-P checklist as a Supplementary File. The main IPD-NMA results paper will follow PRISMA-IPD and PRISMA-NMA statements. We have corrected this information in the revised manuscript (p.6).

5. The PROSPERO registration number should be provided in the protocol and its abstract.

Response: The registration number has now been included in the abstract (p.2), and in the methods section (CRD42019136640) (p.6).

6. The authors mention that they will search certain databases to identify relevant studies. However, in the search codes available in the supplementary, the search dates are already available. Have the searches already been completed? I am also unclear about this when having a look at the network plots. Do these include studies provided in a previous search or the current search and these will be updated?

Response: As recommended in the Cochrane Handbook for Systematic Reviews of Interventions, we have conducted preliminary searches and conducted some preliminary variable mapping before deciding on which variables to collect. For this purpose, the search has been conducted previously, and will be updated upon publication of this protocol to include all relevant studies to date.

We have included the following statement in the revised manuscript:

'Preliminary searches using this search strategy have already been completed, but the search will be updated regularly to include additional trials.' (p.8)

7. Methods - "Women and babies will be included regardless of whether mode of delivery was vaginal or caesarean, and whether the birth was singleton or multiple": How will the authors treat babies born from the same mother in their analysis? Will they assume that they are independent with different characteristics?

Response: Thank you for this excellent point, we have added the following sentence to our revised manuscript: 'Correlations between multiples will be accounted for in the analyses.' (p.6)

8. What data will the authors request from trialists to provide them? The specific contact process needs to be described in the paper (for more details on this see:

<https://scanmail.trustwave.com/?c=13000&d=hd2J3gypC-RFnJlcPGSaRhU5m295fmb5kFPEC3NkcQ&u=https%3a%2f%2fwww%2encbi%2enlm%2enih%2egov%2fpubmed%2f27116943%29>

Response:

We have provided the following additional information in the revised manuscript:

'Each participating trial will be asked to provide de-identified, individual participant level data. Clear instructions will be provided on which data are needed and the secure data transfer process. The preferred data format and coding for each variable will be supplied to the investigators, but data in any format that is most convenient will be accepted and recoded if necessary.' (p.8)

9. "These data will be backed-up and stored in a centralised secure database": More details are required for the data storage and protection. How long will these data be kept in this secured database? Will it be destroyed after few years? Who will have access to these data? Where will the secured database be located at?

Response: As requested by the reviewer, we have provided additional information on data receipt and storage in the revised manuscript. The section now reads as follows:

'Data management will comply with the University of Sydney Data Management Policy 2014, and has been approved by the University of Sydney Human Research Ethics Committee (2018/886).

Depending on trialists' preference, data transfer will either take place via secure data transfer platforms, or shared via institutional secure email using password-protected zip-files. Data for this project will be stored in perpetuity in a password-protected folder within the NHMRC Clinical Trials Centre's network. Only authorised project team members working within the NHMRC Clinical Trials Centre will have access to these data.' (pp.8-9)

10. Details on the process of checking the available data and of producing a consistent format for all trials is missing.

Response: Thank you for this point. The requested information is provided in the manuscript as follows:

'Data checking: For each trial, range and internal consistency of all variables will be checked. Intervention details and missing data will be checked against any protocols, published reports, and data collection sheets. Integrity of the randomisation process will be assessed by examining the chronological randomisation sequence and the balance of participant characteristics across intervention groups. Any inconsistencies or missing data will be discussed with the trialists and resolved by consensus. Each included study will be analysed separately and the results sent to the trial investigators for verification prior to inclusion in the iCOMP database. All trial-specific outcomes generated from the IPD will be cross-checked against published information via a series of crosstabs. Data re-coding: Outcome data may have been collected in different formats across trials. Therefore, the de-identified data from each of the trials will be extracted and re-formatted into a commonly coded dataset.

Data transformation and collating: Once the data from each of the trials are finalised, they will be combined into a single dataset, but a trial identifier code for each participant will be retained. New variables will be generated from the combined dataset as required to address the hypotheses to be tested.' (p.9)

11. Which RoB tool will be used to assess study quality? Is it the newer ROB 2 tool? How results will be categorized? High, Low, and Unclear? In such a case, will the unclear cases be clarified with the original study authors of the IPD?

Response: We will be following the latest Cochrane handbook guidance and therefore will use the new RoB tool (RoB 2). We have added the following statement to our revised manuscript: 'Uncertainties [in risk of bias assessments] will be resolved where possible by contacting study authors.' (p.9)

12. "Analyses will be conducted using the open-source software R": Which R packages will be used to conduct the IPD meta-analysis and network meta-analysis? These should be specified.

Response: This is a large IPD meta-analysis, and therefore, we expect the data processing to take a long time. IPD-NMA are an emerging methodology, with many new packages under development. We would prefer to not specify an analysis package at this stage, to be able to use the latest/ most appropriate packages for our analyses. R packages will be specified in the resulting main results publication.

13. More details on the models to be used are required. Will the authors use a fixed or random intercept and slope? Which heterogeneity method will the authors use to estimate the between-study variance?

Response: As specified in the manuscript, we will be using a random effects model for the network metaanalysis: 'We will calculate a two-step random-effects contrast-based network meta-regression' (p.11).

We added the following sentence to the methods for the network meta-analysis: 'Statistical heterogeneity will be assessed using the I^2 statistic'. (p.11)

14. "In advance of conducting the analyses, we will decide whether there are sufficient reliable data to allow meaningful analysis of any individual outcome": This is unclear - how will this be decided?

Response: From our experience with previous IPD meta-analyses, in some instances data quality is too low to perform meaningful analyses. For instance, in some cases studies planned to collect a variable, but in practice were unable to collect this variable for the majority of participants. Yet, we agree that the sentence above is unclear, and we recognise that cases like this are already covered in the risk of bias assessment and planned sensitivity analyses. We therefore decided to delete this sentence from the manuscript.

15. "all analyses will be specified a-priori in a full Statistical Analysis Plan": I disagree with this. The statistical analysis plan should be defined in this protocol. All potential approaches should be clarified at this point, well before the data is obtained, and the statistical analysis plan should not be data driven.

Response: We agree that all potential approaches need to be pre-specified and made publicly available, and therefore we have ensured that all key elements of our study are pre-specified in the present protocol. A statistical analysis plan will not introduce any new analyses, but will merely elaborate on the key elements included in this protocol.

In previous IPD meta-analyses, we have found it immensely beneficial to create a step-by-step detailed statistical analysis plan, once the collaboration has been fully formed, and before any data is analysed. This plan will then be agreed on and signed off by the collaboration to ensure all analyses

are pre-specified and pre-agreed by the collaboration. The statistical analysis plan will be made publicly available (as an attachment to the registration record) and will not be data driven since it will be finalised before any analysis takes place, and it will not introduce any new analyses that have not already been specified in this protocol.

16. age 13, lines 17-20: Please elaborate on what you mean by "if data permits".

Response: By 'if data permits' we mean if these outcomes were collected by the included studies. We agree that this statement may be confusing and have therefore removed it from the revised manuscript.

17. "Should models not converge, a Bayesian approach will be used instead" This is unclear. What is meant by the models do not converge in a frequentist framework?

Response: In some instances, statistical programs are unable to derive model parameters that fit the data – this is what we describe as models not converging. One reason for this may be insufficient data/information.

In a Bayesian framework there are pre-specified priors, which result in more information being available compared to a frequentist framework, and therefore a higher likelihood that models may converge. For this reason if a model does not converge in a frequentist framework we plan to use the Bayesian approach.

18. setting a prior of no effect and a large variance" The specific priors that will be used in the Bayesian models should be provided. Also, details on the models to be applied and the software to be used should be specified at the protocol stage.

Response: Thank you for this comment. We have specified a weakly-informative prior in the revised manuscript ($N(0, 5)$) (p.11), assuming log RR between minus 10 and 10. As outlined above, we will be using R for all analyses (as specified in the manuscript), but will not specify packages at the current stage.

19. "potential known sources for inconsistency will be explored" The potential effect modifiers should be specified a priori and at the protocol stage.

Response: Thank you for this comment. We agree that covariates should be specified a priori. In the revised manuscript we now do so as follows: 'Any detected inconsistency will be explored by including covariates specified above (gestational age at birth, hospital setting, as well as study year) into the model, and by excluding potential outlier studies in sensitivity analyses.' (pp.11-12)

20. "A judgement of excessive heterogeneity or inconsistency would prevent the interpretation and reporting of the network meta-analysis." Please clarify this sentence.

Response:

Thank you for this comment. We have revised the sentence as follows: 'In case of a judgement of excessive heterogeneity or inconsistency we would still report the resulting parameters, but would interpret the results as not reliable.' (p.12)

21. Citation 58 should refer to the loop-specific approach. A relevant citation for the low power of inconsistency tests is: <https://scanmail.trustwave.com/?c=13000&d=hd2J3gypC-RFnJlCPGSaRhU5m295fmb5kAiTCyNkeQ&u=https%3a%2f%2fwww%2encbi%2enlm%2enih%2egov%2fpubmed%2f25239546>

Response: Thank you for pointing us to this citation, we have included it in the revised manuscript.

22. We expect to also be able to include some unreported outcomes sourced from the individual participant data provided by the included studies, alleviating selective outcome reporting bias." Is there published evidence for this? Can you please provide relevant citations to back-up this statement?

Response: This advantage of IPD meta analyses has been listed in the latest Cochrane handbook chapter on IPD meta-analyses, and we now cite this chapter in our revised manuscript. (p.12)

In addition, our team has conducted over 15 individual participant data meta-analyses to date, and from our experience we frequently were able to include additional outcomes that had not been reported by the study authors. Nonetheless, we have toned down this statement in the revised manuscript as follows: 'We expect to also be able to include some unreported outcomes sourced from the individual participant data provided by the included studies, which may alleviate selective outcome reporting bias.' (p.12)

23. "For other secondary outcomes, no formal adjustments for multiple testing are planned but instead, we will be following the approach outlined by Schulz and Grimes(60):" Please clarify what is meant by multiple testing. Also, I believe the reader will not be familiar with the Schulz and Grimes(60) approach. Please provide more details on this.

Response: We have added the following information to the revised manuscript, to clarify this issue:

'For other secondary outcomes, no formal adjustments for multiplicity (i.e. the accumulation of type 1 error and thus higher likelihood of chance findings when assessing multiple outcomes) are planned. Instead, we will be taking the following approach outlined by Schulz and Grimes(60): as secondary outcomes examined in this study are interrelated, we will interpret the pattern of results, examining consistency of results across related outcomes, instead of focusing on any single, statistically significant result.' (p.12)

24. The authors should specify for which outcomes they are going to apply subgroup analyses and metaregression analyses.

Response: Thank you for pointing this out, we have specified the outcomes for subgroup analyses in the revised manuscript. 'Subgroup analyses will be conducted for the primary outcome (death before discharge), and the two key secondary outcomes (IVH any grade, blood transfusion).' (p.11)

25. Discussion: A paragraph outlining the limitations of the present study is missing.

Response: Thank you for this comment. As is usual practice, a limitations paragraph will be added to the main results manuscript, not to the protocol. A brief overview of limitations is provided in the 'Strengths and limitations of this study' section. (p.3)

26. Patient and Public Involvement: Please clarify how the patient comments will be included in this project.

Response: Thank you for this excellent point. We have invited two consumer representatives to join the iCOMP collaboration, comment on this protocol and be involved in the interpretation of results. We have added a paragraph 'Public and patient involvement' outlining this to the protocol. (p.13)

VERSION 2 – REVIEW

REVIEWER	Michael P Meyer Neonatal Unit, Middlemore Hospital Auckland New Zealand
REVIEW RETURNED	29-Jan-2020

GENERAL COMMENTS	Thank you for your response.
------------------------------

REVIEWER	Stefano Ghirardello, MD Fondazione IRCCS Ca' Granda Ospedale Maggiore Policlinico, Milan, Italy
REVIEW RETURNED	30-Jan-2020

GENERAL COMMENTS	Thanks for having provided the requested revisions.
---

REVIEWER	Areti Angeliki Veroniki University of Ioannina, Greece
REVIEW RETURNED	09-Feb-2020

GENERAL COMMENTS	The authors have satisfactorily revised the manuscript and it is ready for publication. I have no additional comments.
--